# Inter-Observer Agreement between Low-Dose and Standard-Dose CT with Soft and Sharp Convolution Kernels in COVID-19 Pneumonia

**DOI:** 10.3390/jcm11030669

**Published:** 2022-01-27

**Authors:** Ivan Blokhin, Victor Gombolevskiy, Valeria Chernina, Maxim Gusev, Pavel Gelezhe, Olga Aleshina, Alexander Nikolaev, Nicholas Kulberg, Sergey Morozov, Roman Reshetnikov

**Affiliations:** 1Research and Practical Clinical Center for Diagnostics and Telemedicine Technologies of the Moscow Health Care Department, 127051 Moscow, Russia; chernina909@gmail.com (V.C.); gelezhe.pavel@gmail.com (P.G.); a.e.nikolaev@yandex.ru (A.N.); morozov@npcmr.ru (S.M.); reshetnikov@fbb.msu.ru (R.R.); 2Artificial Intelligence Research Institute (AIRI), 105064 Moscow, Russia; g_victor@mail.ru; 3Moscow Polytechnic University, 107023 Moscow, Russia; maxtox3@gmail.com; 4City Clinical Hospital No. 13 of the Moscow Health Care Department, 115280 Moscow, Russia; olya.aleshina.tula@gmail.com; 5Federal Research Center “Computer Science and Control” of Russian Academy of Sciences, 119333 Moscow, Russia; kulberg@yandex.ru; 6Institute of Molecular Medicine, I.M. Sechenov First Moscow State Medical University, 119991 Moscow, Russia

**Keywords:** COVID-19, SARS-CoV-2, pneumonia, tomography, X-ray computed

## Abstract

Computed tomography (CT) has been an essential diagnostic tool during the COVID-19 pandemic. The study aimed to develop an optimal CT protocol in terms of safety and reliability. For this, we assessed the inter-observer agreement between CT and low-dose CT (LDCT) with soft and sharp kernels using a semi-quantitative severity scale in a prospective study (Moscow, Russia). Two consecutive scans with CT and LDCT were performed in a single visit. Reading was performed by ten radiologists with 3–25 years’ experience. The study included 230 patients, and statistical analysis showed LDCT with a sharp kernel as the most reliable protocol (percentage agreement 74.35 ± 43.77%), but its advantage was marginal. There was no significant correlation between radiologists’ experience and average percentage agreement for all four evaluated protocols. Regarding the radiation exposure, CTDI_vol_ was 3.6 ± 0.64 times lower for LDCT. In conclusion, CT and LDCT with soft and sharp reconstructions are equally reliable for COVID-19 reporting using the “CT 0-4” scale. The LDCT protocol allows for a significant decrease in radiation exposure but may be restricted by body mass index.

## 1. Introduction

The World Health Organization (WHO) declared a public health emergency on 30 January 2020 due to coronavirus disease 2019 (COVID-19) [1]. While the value of medical imaging in COVID-19 diagnostics remains uncertain [2], computed tomography (CT) is a primary imaging modality used to assess the disease severity in cases of suspected or confirmed COVID-19 pneumonia [3,4,5].

Studies evaluating CT applicability for COVID-19 indicate the absence of pathognomonic signs but highlight the frequently occurring findings: bilateral peripheral ground-glass opacities and “crazy-paving” pattern with basal predominance [6,7]. The association between the CT findings and COVID-19 depends on the local prevalence of the disease and could be influenced by individual factors such as exposure history [8]. However, CT is widely used for the evaluation of lung parenchyma [9]. Several CT severity score systems for the quantitative assessment of pulmonary lesions in COVID-19 have been described. Yang et al. divided 18 segments of both lungs into 20 regions, subjectively evaluating lung opacities with scores of 0, 1, and 2 for 0%, <50%, or ≥50% involvement of each region, respectively. The final score is the sum of the individual scores in the 20 regions, ranging from 0 to 40 [10]. Salaffi et al. used a system with two scores, “extent” and “nature”, and three evaluation zones for each lung. “Extent” scoring assigns 0 for normal lung, 1 for <25% lung abnormality, 2 for 25–49%, 3 for 50–74% and 4 for ≥75% abnormality. “Nature” scoring reflects the type of pulmonary involvement with 1 for normal lung, 2 for at least 75% ground-glass or crazy paving, 3 for ground-glass, crazy paving and consolidation with less than 75% involvement each, and 4 for at least 75% consolidation; the extent and nature scores are then multiplied by each other and summed at all six levels; the final radiological severity score ranges from 0 to 96 [11]. The latter was initially used by the Russian Federation, but due to its complexity the alternative visual semi-quantitative grading system “CT0-4” was developed and adopted (CT0 category corresponds to 0% lung involvement, CT1: <25%, CT2: 25–50%, CT3: 50–75%, and CT4: >75% involvement) [12,13,14].

While there is a degree of standardization in assessing COVID-19 pneumonia severity, no normalization is currently applied for the scanning protocol. According to the ALARA principle (As Low As Reasonably Achievable), it is recommended that the lowest possible dose be used, while maintaining the diagnostic quality for the clinical task at hand [15]. WHO issued recommendations on COVID-19 imaging, advocating the usage of a low-dose CT in adults with the disease [16]. Several studies confirm the applicability of low- and ultra-low-dose chest CT protocols for detecting typical COVID-19 pneumonia signs [17,18]. In practice, the effective dose given to COVID-19 patients may vary widely depending on the medical center, and it is generally higher than for patients with other pulmonary infectious diseases [19]. There is also uncertainty about what reconstruction kernel to use for the interpretation of CT images in COVID-19, both for human readers and artificial intelligence. Previous studies demonstrate that kernel selection affects pulmonary nodule measurement in routine [20] and low-dose CT [21], as well as emphysema quantification [22]. A general recommendation for chest CT is to use both soft and sharp kernels [23]. This recommendation also applies to studies in machine learning [24] and radiomics [25]. However, Kwee et al. stated that CT in COVID-19 should be viewed using only a sharp kernel [26].

The study aimed to develop an optimal COVID-19 CT scanning protocol in terms of safety and reliability. For that, we assessed the inter-observer agreement between standard- and low-dose CT with soft and sharp kernels using the CT0-4 severity score.

## 2. Hypotheses

Two hypotheses were put forward to achieve the study objective:The CT0-4 grades assigned by readers for the same patient do not differ between CT and LDCT protocols reconstructed with the sharp or soft kernels;The inter-rater reproducibility of CT0-4 grades assigned by radiologists of different levels of experience does not depend on the kernel and protocol selection.

## 3. Materials and Methods

Ethical approval No. 3/2020 was granted by the regional ethics board, Independent Ethics Committee, Moscow Regional Office of the Russian Society of Radiologists and Radiographers. All participants completed and signed a dedicated consent form before study participation. The study was registered at ClinicalTrials.gov, ID NCT04379531, date of registration 7 May 2020.

### 3.1. Case Selection

This multicenter study was conducted in two outpatient clinics in Moscow, Russia. We included a consecutive sample of patients aged ≥18 years referred for a chest CT scan by their physician between 6 May 2020, and 22 May 2020, due to suspected viral pneumonia. The clinical symptoms warranting increased suspicion were as follows: increased body temperature (>37 °C); cough without sputum or with a small amount of clear sputum; and increased respiratory rate (>22 breaths per minute). We excluded pregnant and lactating women from the study, as well as patients with implanted foreign objects at the scan level (including cardiac pacemakers and metal constructions); patients with a history of thoracic surgery; cancer patients; and patients refusing to participate in the study.

During a single visit to the CT room, each patient underwent two consecutive scans with standard- and low-dose CT protocols at peak inspiration depth. The scans were acquired consecutively with subjects in the supine position, from the diaphragm to the apex of the lung within a single breath-hold. The patient was not removed from the table between the scans. We did not implement respiratory inductance plethysmography to ensure similar lung volume between the acquisitions due to high patient flow.

### 3.2. CT Acquisition

Chest CT was performed using a 64-detector CT scanner (Aquilion 64, Canon, Japan) without iterative reconstruction algorithms. We used the vendor-provided standard chest CT protocol and had previously developed a low-dose CT protocol for COVID-19 [27].

For both standard- and low-dose CT scans, all data acquisition settings were the same, except for the automatic exposure control (Sure Exposure 3D):(1)For a standard CT scan, the tube current was automatically adjusted over the entire scan length from 40 to 500 mA, with a noise level of 10 (standard deviation) for 5.0 mm slices.(2)For a low-dose CT scan, the current was automatically adjusted over the entire scan length from 10 to 500 mA, with a noise level of 36 for 5.0 mm slices.

Additional CT scan parameters were the same for standard- and low-dose CT scans: voltage 120 kV; rotation time 0.5 s; direction—out (from legs to head); modulation XY—on; collimation 64 × 0.5 mm; helical pitch 53.0; and scan time—6 s on average. An intravenous contrast agent was not used.

Images were reconstructed as sharp (FC51) and soft (FC07) series with a 1 mm slice thickness for both standard- and low-dose scans.

### 3.3. Image Analysis

Reading of CT scans was performed by ten radiologists with 3–25 years’ experience, and trained in COVID-19 pneumonia interpretation. The FAnTom software version 1.2 (Moscow, Russia) [28,29] was modified to assess disease severity according to the CT0-4 scoring scheme (Figure 1); it was also used to facilitate annotation and provide the radiologists with online access to a randomly selected set of anonymized studies.

Each series was independently evaluated by two raters (Reader A and Reader B) blinded to the protocol used (standard- or low-dose CT, sharp or soft kernel) through the FAnTom interface.

### 3.4. Sample Size Consideration

The sample size estimation was made according to Walter et al. [30]. We assumed 0.6 as the minimally acceptable level of correlation between the CT0-4 grades assigned by readers, and 0.7 as a value of expected inter-rater correlation for each of the four protocols. Every case was assessed by two readers; in order to achieve 80% statistical power at a 5% significance level, the study required 205.4 subjects. To account for the multicenter character of the study, we increased this number by 10%, resulting in 230 participants.

### 3.5. Statistical Analysis

For statistical analysis, data from all subjects included in the study (Full Analysis Set, FAS) were used.

The following characteristics were calculated to describe data representing numeric variables (such as age or bodyweight): number of non-missing values; arithmetic mean; and standard deviation.

For qualitative and category variables (sex, CT0-4 grade), the percentage of subjects in each category is provided.

Inter-observer agreement in CT0-4 grades for standard-dose CT with sharp kernel FC51 (CT sharp); standard-dose CT with soft kernel FC07 (CT soft); low-dose CT with sharp kernel FC51 (LDCT sharp); and low-dose CT with soft kernel FC07 (LDCT soft) was analyzed using percent agreement (PA) and Cohen’s kappa metrics.

For testing of Hypothesis 1, we compared paired data on CT0-4 scoring for the same patient across the CT sharp, CT soft, LDCT sharp, and LDCT soft protocols. Since for each of the four protocols, there were only two data points per patient (assessments by Reader A and Reader B), the data were unsuitable for Cohen’s kappa calculations. For that reason, we tested Hypothesis 1 using only the percentage agreement metric. The data were divided into four matched subsets corresponding to the protocols used, and the PA values between readers A and B were calculated for each case within each subset. These individual agreements were used to calculate the mean PAs for each protocol and compare the protocols using the Wilcoxon signed-rank test [31]. The correlation between the CT0-4 grades assigned by the readers for the same patient was also assessed using Spearman’s rank correlation coefficient.

For testing of Hypothesis 2, we collected agreement data for each pair of readers within the four subsets. A pair of readers was included in the analysis if the pair interpreted ≥3 cases. For every reader, their overall agreement statistic was calculated from pair agreements. These values were then used to calculate mean inter-observer agreement values for each of the four protocols and compare the agreement metrics between the protocols using the Wilcoxon signed-rank test [31].

We also quantified the agreement using the differences between the scores assigned by readers. The difference statistics were compared between the four protocols using the Wilcoxon signed-rank test.

All distributions were checked for normality using a Q–Q plot and the Shapiro–Wilk’s W test [32].

All comparisons were made at a statistical significance level of 0.05. For statistical analyses, the Stata 14 program (StataCorp, College Station, TX, USA) and dplyr [33], ggplot [34], and irr [35] packages for R 3.6.3 (Vienna, Austria) [36] were used.

## 4. Results

The total sample size was 230 patients (55.6% females), with the following averages: age 47 ± 15 years, weight 80 ± 18 kg, height 169 ± 10 cm, and body mass index 27.9 ± 5.6 kg/m^2^. The distribution of COVID-19 severity within the sample was 45%, 32%, 17%, 5%, and 1% for CT0, CT1, CT2, CT3, and CT4 categories, respectively.

For the standard CT protocol, the average CT dose index (CTDI_vol_) was 14 ± 4 mGy and the average dose-length product (DLP) was 591 ± 192 mGy·cm, with an average effective dose of 10 ± 3 mSv.

For the low-dose CT protocol, the average CTDI_vol_ was 4 ± 3 mGy and the average DLP was 184 ± 118 mGy*cm, with an average effective dose of 3 ± 2 mSv.

For the low-dose protocol, CTDI_vol_ was 3.6 ± 0.64 times lower compared with the standard CT protocol.

Each reader independently interpreted 184 randomly assigned cases. On average, each pair of readers assessed 7.3 ± 1.1 cases of each protocol.

### 4.1. Hypotheses Findings

**Hypothesis** **1** **(H1).**The CT0-4 grades assigned by readers to the same patient do not differ between CT and LDCT protocols reconstructed with the sharp or soft kernels (kernel and protocol selection do not affect reporting in COVID-19).

Each case was presented to the readers in one of four reconstructions (Figure 2). The highest inter-observer agreement was observed for the LDCT sharp protocol, while the LDCT soft protocol had the lowest PA value (Table 1). These two protocols were the only pair with a statistically significant difference for which the Wilcoxon rank-sum test rejected Hypothesis 1 (Table 1).

**Hypothesis** **2** **(H2).***The inter-rater reproducibility of CT0-4 grades assigned by radiologists of different levels of experience does not depend on the kernel and protocol selection*.

There was no significant correlation between the level of a radiologist’s experience and their average PA value for the CT sharp (rho = −0.01, *p* = 0.97), CT soft (rho = −0.43, *p* = 0.22), LDCT sharp (rho = −0.13, *p* = 0.72), and LDCT soft (rho = −0.24, *p* = 0.49) protocols. Note that for the soft kernel, the Spearman’s correlation coefficients had lower negative values, suggesting that experienced radiologists were slightly (but not significantly) more likely to disagree with their partner when using this reconstruction method.

Following the results obtained for Hypothesis 1 testing, the protocols using sharp kernels had a higher average inter-rater agreement in comparison with the soft kernel protocols, both in PA and Cohen’s kappa values (Table 2). However, this difference was not statistically significant according to the Wilcoxon test results (Table 2).

When analyzing the results, we discovered that Cohen’s kappa was not an optimal metric to quantify the inter-rater agreement in CT0-4 grades. For example, there were hard-to-interpret differences between the percent agreement and Cohen’s kappa values for some pairs of readers (Appendix A). The difference is reflected in the *p*-values when comparing the percentage agreements or Cohen’s kappa values across the protocols (Table 2).

### 4.2. Differences between the “CT0-4” Scores Assigned by Readers

To further evaluate the inter-rater agreement, we studied the differences between the CT0-4 grades assigned by the readers. For all four protocols, the difference distribution was non-normal with positive kurtosis. Most differences (69.6 ± 4.2%) were zero, while 27.6 ± 3.8%, 2.3 ± 0.5%, 0.4 ± 0.6%, and 0.2 ± 0.3% of assessments differed by one, two, three, and four grades, respectively (Figure 3).

In agreement with the results obtained for Hypotheses 1 and 2, the most reliable protocol was LDCT sharp, and the protocol with the most mismatches was LDCT soft (Figure 3). While the difference between the protocols was not statistically significant, we find this observation interesting as it highlights again the marginal advantage of the LDCT sharp protocol over the alternatives.

## 5. Discussion

We assessed the inter-observer agreement between standard- and low-dose CT with soft and sharp kernels using the CT0-4 severity score. The most reliable protocol in terms of percentage agreement was LDCT reconstructed with a sharp kernel, and the protocol with the most mismatches was LDCT reconstructed with a soft kennel. These two protocols were the only pair with statistically significant differences for inter-observer agreement. There was no significant correlation between the level of radiologists’ experience and their average percentage agreement value for all four evaluated protocols. Regarding the radiation exposure, CTDI_vol_ was 3.6 ± 0.64 times lower for the low-dose protocol, compared with the standard CT protocol.

The proposed LDCT protocol for COVID-19 reconstructed with soft and sharp kernels is associated with a significant decrease in the effective dose and is unlikely to negatively impact the accuracy of LDCT reporting in COVID-19. A similar conclusion was reached by Samir et al. in a prospective study comparing ultra-low-dose CT with a 22 mAs tube current and low-dose CT. The former had excellent inter-observer agreement (96–100%) and 90.38–93.84% accuracy in COVID-19 imaging [18]. Tabatabaei et al. obtained similar results on a small sample of 20 patients with an intraclass correlation of 0.98–0.99 for detecting COVID-19 pneumonia using a 30 mAs low-dose chest CT protocol (effective dose of 1.80 ± 0.42 mSv) [17]. Note that both studies used binary classification for detecting signs of pneumonia, but without severity assessment. To our knowledge, no prior studies have compared the standard- and low-dose CT protocols with different convolution kernels using a semi-quantitative scale for lung lesions in suspected COVID-19.

Regarding the convolution kernel choice, our findings agree with the recommendations of Kwee et al., suggesting usage of the sharp kernel for pulmonary tissue evaluation in COVID-19 patients [26]. The marginal advantage of the combination of the low-dose protocol and sharp convolution kernel is also consistent with the Fleischner Society guidelines for pulmonary nodule management [20,37].

Image noise is one of the main factors determining the diagnostic performance of CT, as it has a direct effect on image quality and, therefore, on the radiation exposure level specified for a study. Overweight and obese patients (BMI ≥ 25) often suffer from reduced CT image quality at a typical effective dose due to higher noise levels. It is particularly pertinent in chest CT, especially when implementing specialized scanning protocols [38,39].

The data on chest LDCT in overweight patients are contradictory. For example, according to Ohana et al., LDCT has limited potential in comparison with standard CT in obese patients (BMI > 35) with interstitial pneumonia [40]. These findings restrict the applicability of the *LDCT sharp* protocol to the subjects with a normal BMI.

In addition to gold-standard imaging (CT and LDCT), chest radiography (CXR), lung ultrasound (LUS), and magnetic resonance imaging (MRI) can also be used for COVID-19-associated pneumonia detection [41,42]. LUS has emerged as a radiation-free alternative to CT and LDCT, but its practical applicability is controversial. Some studies show a high level of diagnostic agreement between LUS and CT [43], while others point to its subjectivity and operator dependence [44], restricting LUS usage to critically ill patients, pregnant women, children, and the bedridden aged population [45]. CXR is readily available and operator independent. Ghosh et al. suggest CXR as the initial imaging modality in every suspected case of COVID-19 regardless of laboratory status [42]. Finally, several studies show chest MRI to be promising in COVID-19-associated pneumonia diagnosis and severity assessment [46,47], but its limitations include poor availability, longer examination times, and high susceptibility of the MRI equipment to contamination [48].

Our study has several limitations. First, the study was performed on one CT scanner model, as it is the most common device in the equipment parks at Moscow City Health Department medical organizations. The recommended protocols for other models and manufacturers are likely to differ from that proposed. Second, we did not compare the regional CT0-4 system with other generally accepted semi-quantitative scales [10,11,49]. Third, we did not evaluate the feasibility of noise suppression with iterative reconstructions, neural networks or X-ray beam filtration. The use of a tin filter is not yet widespread but allows a nearly 90% effective dose reduction in chest CT in suspected COVID-19 by filtering the X-ray beam [50]. The increasing prevalence of units with iterative reconstruction or a tin filter may serve as a topic for future research. Fourth, we did not implement respiratory inductance plethysmography to ensure similar lung volume between the acquisitions.

## 6. Conclusions

Standard CT and LDCT protocols with soft and sharp image reconstructions are equally reliable for COVID-19 reporting using the CT0-4 grading system with a marginal advantage of the *LDCT sharp* protocol. The LDCT protocol allows for a significant decrease in radiation exposure, but its applicability may be restricted by the patient’s body mass index.

## Figures and Tables

**Figure 1 jcm-11-00669-f001:**
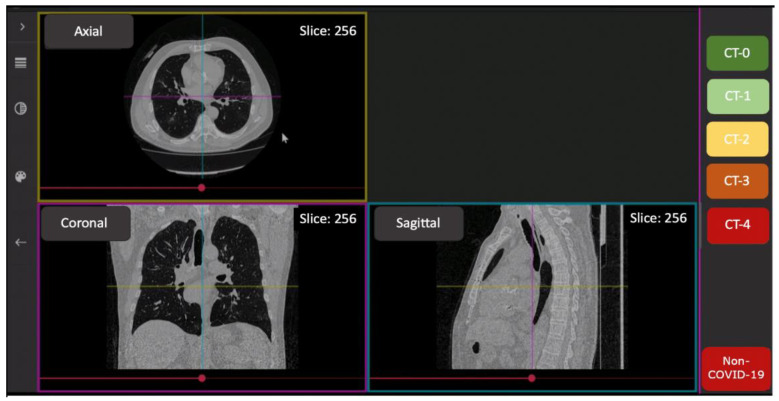
The interface of the FAnTom software tailored for CT0-4 grading.

**Figure 2 jcm-11-00669-f002:**
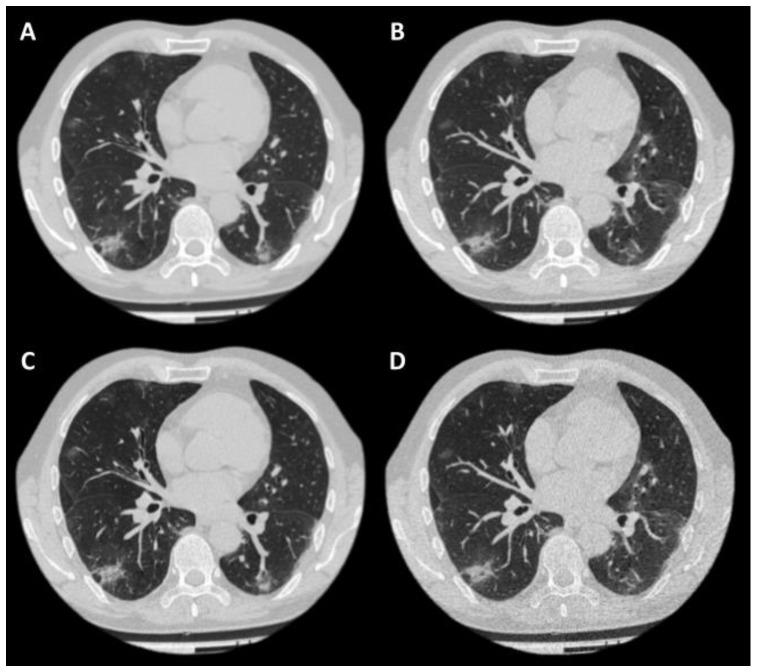
Example of a case in four reconstructions. Patient: 57-year-old male, BMI = 24.05 kg/m^2^; (**A**)—CT soft; (**B**)—LDCT soft; (**C**)—CT sharp; (**D**)—LDCT sharp. CTDI_vol_ for CT was 15.5 mGy, for LDCT—3.4 mGy. The effective dose for standard CT was 10.31 mSv, for LDCT—2.46 mSv.

**Figure 3 jcm-11-00669-f003:**
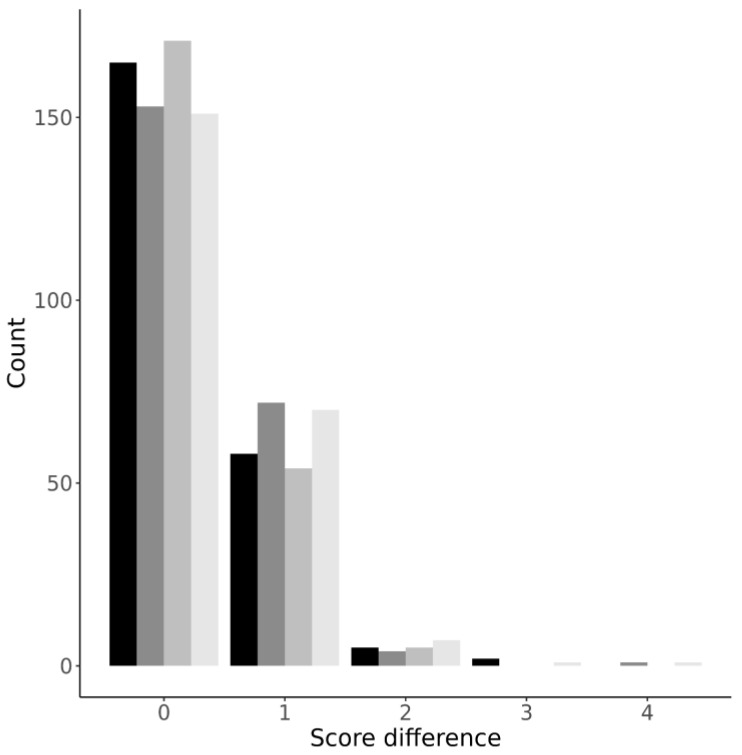
Distribution plot of differences in CT0-4 grades across the four protocols. From darker to lighter: LDCT sharp; LDCT soft; CT sharp; CT soft.

**Table 1 jcm-11-00669-t001:** Inter-observer agreement for matched patient data.

	CT Sharp		CT Soft		LDCT Sharp		LDCT Soft	
PA, %	71.74 ± 45.12		66.52 ± 47.29		74.35 ± 43.77		65.65 ± 47.59	
Spearman’s *rho*	0.78, *p* < 0.001		0.77, *p* < 0.001		0.82, *p* < 0.001		0.75, *p* < 0.001	
*p*-value		0.23		0.07		0.04 *		
		0.53						
				0.84				
		0.16						

* Statistically significant difference, *p* < 0.05.

**Table 2 jcm-11-00669-t002:** Average inter-observer agreements for the readers in the study.

Radiologist	Experience, Years	CT Sharp		CT Soft		LDCT Sharp		LDCT Soft	
		PA, %	Cohen’s Kappa	PA, %	Cohen’s Kappa	PA, %	Cohen’s Kappa	PA, %	Cohen’s Kappa
0	3	75.1 ± 13.7	0.59 ± 0.23	82.6 ± 10.8	0.73 ± 0.15	89.3 ± 12.4	0.82 ± 0.21	86.6 ± 13.9	0.79 ± 0.21
1	10	54.0 ± 20.0	0.31 ± 0.29	49.5 ± 14.1	0.29 ± 0.15	63.9 ± 13.1	0.49 ± 0.16	43.6 ± 18.4	0.19 ± 0.19
2	25+	87.8 ± 14.0	0.78 ± 0.26	66.8 ± 31.3	0.55 ± 0.40	82.5 ± 11.2	0.68 ± 0.18	68.2 ± 20.6	0.53 ± 0.28
3	6	70.5 ± 11.1	0.39 ± 0.23	55.5 ± 11.1	0.29 ± 0.20	62.5 ± 21.9	0.38 ± 0.32	59.7 ± 7.5	0.38 ± 0.09
4	2	72.0 ± 18.2	0.49 ± 0.38	74.6 ± 18.7	0.62 ± 0.25	73.4 ± 17.8	0.54 ± 0.3	75.8 ± 19.9	0.65 ± 0.27
5	4	65.6 ± 29.5	0.44 ± 0.50	77.7 ± 15.8	0.55 ± 0.35	62.4 ± 16.5	0.44 ± 0.19	77.1 ± 19.4	0.65 ± 0.27
6	1	74.1 ± 17.4	0.51 ± 0.28	72.0 ± 31.3	0.62 ± 0.40	73.1 ± 16.4	0.56 ± 0.31	54.4 ± 29.2	0.36 ± 0.39
7	7	77.5 ± 6.2	0.64 ± 0.10	64.8 ± 16.2	0.44 ± 0.28	75.4 ± 27.7	0.62 ± 0.42	75.0 ± 18.4	0.61 ± 0.26
8	2	78.4 ± 17.8	0.53 ± 0.38	58.9 ± 11.9	0.44 ± 0.28	82.6 ± 14.4	0.7 ± 0.21	68.5 ± 20.0	0.58 ± 0.29
9	6	73.6 ± 21.1	0.57 ± 0.34	60.5 ± 26.9	0.34 ± 0.35	62.5 ± 17.4	0.44 ± 0.16	63.4 ± 19.1	0.46 ± 0.24
Average		72.9 ± 8.8	0.53 ± 0.13	66.3 ± 10.5	0.49 ± 0.15	72.8 ± 9.8	0.57 ± 0.14	67.3 ± 12.5	0.52 ± 0.17
*p*-value, PA/Cohen’s kappa			0.19/0.58		0.25/0.31		0.48/0.58		
			0.85/0.68						
					0.79/0.58				
			0.25/0.79						

## Data Availability

The data presented in this study are available on request from the corresponding author. The data are not publicly available due to regional legislative policies.

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
