# Peer review of "Inter-Observer Agreement between Low-Dose and Standard-Dose CT with Soft and Sharp Convolution Kernels in COVID-19 Pneumonia"

_jcm, 2022, doi:10.3390/jcm11030669_

Round 1

Reviewer 1 Report

Just a minor point or reflection submitted to the Authors: their approach to the diagnosis of COVID-19 pneumonia is syntonic with the one adopted by ILO-BIT for the detection and characterization of pneumoconiotic radiological patterns and with the correspondent one adopted for the ICOERD classification. In my humble opinion, very interesting the fact that similar criteria guide the coping with the common difficulties of the recognition and the classification of lung diseases of quite different origin.     

Author Response

The authors thank the reviewer for their insight. There are indeed CT findings that could be associated with a number of different pathologies. For example, ground glass opacities can be observed in opportunistic and non-opportunistic infections, chronic interstitial diseases, acute alveolar diseases, and other cases. To reflect upon that, we have revised p.2 of the Introduction (lines 38-41).

Reviewer 2 Report

An important paper

Author Response

We thank the reviewer for the positive evaluation of our work.

Reviewer 3 Report

The manuscript is an important clinical document for the early management of the patient with SARS-CoV-2 infection. The research design is not original, many scientific articles in the literature deal with the same topic. 

Author Response

The authors thank the referee for the valuable comments. We carefully revised the manuscript and verified the data analysis. It turned out that we inappropriately used Pearson’s correlation analysis because of the non-parametric character of our data. In the revised text we used Spearman’s rank correlation coefficient to describe the association between the variables (Methods, line 162; Results, Table 1 and lines 212-214). To further improve the quality of the manuscript, we have revised Figure 2 and computed p-values for Cohen’s kappa statistics (Table 2).

We would like to point out that to our knowledge, there are no study designs comparing interpretations of CT and LDCT data in COVID-19 for the same patient in the published literature. This determines the novelty of our work and the importance of its results.